# Mental Health and Lifestyle Factors Among Higher Education Students: A Cross-Sectional Study

**DOI:** 10.3390/bs15030253

**Published:** 2025-02-23

**Authors:** Raquel Simões de Almeida, Andreia Rodrigues, Sofia Tavares, João F. Barreto, António Marques, Maria João Trigueiro

**Affiliations:** 1LabRP-CIR, E2S, Polytechnic of Porto, 4200-135 Porto, Portugal; 10200797@ess.ipp.pt (A.R.); 10200727@ess.ipp.pt (S.T.); jbarreto@ess.ipp.pt (J.F.B.); ajmarques@ess.ipp.pt (A.M.); mjtrigueiro@ess.ipp.pt (M.J.T.); 2CPUP, University of Porto, 4200-135 Porto, Portugal

**Keywords:** mental health, lifestyle, higher education, young people, well-being

## Abstract

This study aimed to describe the lifestyle factors and mental health levels among higher education students and identify their predictors. A cross-sectional study with a sample of 745 students was conducted with students from the Polytechnic of Porto using the Depression Anxiety Stress Scales (DASS-21), Clinical Outcomes in Routine Evaluation (CORE)-18, and FANTASTICO Lifestyle Questionnaire. The findings indicate that while students generally exhibited a positive lifestyle, they also experienced mild levels of depression, anxiety, and stress, nearing the moderate threshold. The DASS-21 Depression subscale was a significant predictor of both CORE-18 and FANTASTICO scores, underscoring the strong relationship between depression and overall well-being. Anxiety and stress were also predictors of CORE-18 scores, reflecting the negative impact of stress on students’ psychological well-being. Perceived health status and the male sex were associated with better outcomes on the DASS-21 and CORE-18, while the female sex predicted a healthier lifestyle, as measured by FANTASTICO. These findings highlight the importance of targeted interventions that address mental health and promote healthy lifestyle choices in educational settings.

## 1. Introduction

The mental health of higher education students has gained increasing attention in recent years, as a growing body of research reveals concerning trends in the prevalence of mental health issues such as stress, anxiety, and depression within this population ([12]; [13]; [63]).

The transition to university life represents a critical period for many young adults, marked by significant changes and challenges that can impact their psychological well-being ([88]). The demands of academic performance, the need to establish new social connections, the challenges of independent living and learning, and the struggle to maintain a balanced lifestyle contribute to the complex mental health landscape faced by students ([7]; [78]).

A key factor in understanding student mental health is the interplay between psychological well-being and lifestyle choices. The [87] ([87]) defines mental health as a state of well-being in which an individual realizes their own abilities, can cope with the normal stresses of life, can work productively, and is able to contribute to their community. This definition underscores the importance of both mental and physical health as interrelated components of overall well-being. Research has consistently shown that lifestyle factors, such as physical activity, sleep quality, nutrition, substance use, and social relationships, are closely linked to mental health outcomes ([18]; [26]; [86]).

Physical activity, for instance, has been extensively studied for its beneficial effects on mental health. Regular exercise is associated with lower levels of stress, anxiety, and depression, as well as improved mood and cognitive function ([52]; [48]). The mechanisms underlying these effects are thought to include the release of endorphins, improved sleep, and the promotion of neurogenesis and brain plasticity. Despite these well-documented benefits, many students struggle to incorporate regular physical activity into their routines, often due to lack of motivation, lack of time, academic pressures, or a lack of access to facilities ([24]).

Sleep quality is another critical lifestyle factor influencing mental health. Sleep disturbances are common among higher education students, with many experiencing irregular sleep patterns, insufficient sleep duration, and poor sleep quality ([5]). These disruptions can lead to a range of negative outcomes, including impaired cognitive function, reduced academic performance, and increased susceptibility to mental health disorders such as depression and anxiety ([79]). The relationship between sleep and mental health is bidirectional, with poor mental health also contributing to sleep problems, creating a vicious cycle that can be difficult to break ([89]).

Nutrition plays a similarly important role in maintaining mental health. A growing body of evidence suggests that dietary patterns can significantly affect mental well-being. Healthy eating patterns, such as the Mediterranean diet, are associated with lower risks of depression and anxiety, while diets high in processed foods, sugars, and unhealthy fats are linked to poorer mental health outcomes ([25]; [92]). Nutrient deficiencies, particularly in omega-3 fatty acids, B vitamins, and magnesium, have also been implicated in the development and exacerbation of mental health issues ([93]).

Social relationships and support networks are crucial for mental health, particularly during the transition to university life. Positive social interactions and a strong sense of community can provide emotional support, reduce feelings of isolation, and buffer against the negative effects of stress ([80]). Conversely, social isolation, loneliness, and poor social integration are significant risk factors for the development of mental health disorders ([10]).

Substance use, including alcohol, tobacco, and recreational drugs, is another important lifestyle factor that can impact mental health. While some students may use substances as a coping mechanism for stress, anxiety, or social pressures, substance use can exacerbate mental health problems and lead to long-term negative consequences ([85]).

Additionally, the escalating climate crisis significantly influences the relationship between mental health and lifestyle factors among higher education students. Extreme weather events, such as heatwaves and rising temperatures, have been linked to increased instances of mood disorders, anxiety, and impaired cognitive functions. For instance, individuals with pre-existing mental health conditions are three times more likely to face fatal outcomes during heatwaves compared to those without such conditions ([17]). Also, exposure to air pollution has been associated with decreased cognitive abilities and a higher prevalence of behavioral issues ([16]).

Given the multifaceted and bidirectional nature of the relationship between mental health and lifestyle choices and associated constraints, it is important to take a comprehensive approach when examining the well-being of higher education students. Also, despite the growing recognition of these issues, a need for more comprehensive research remains if we are to fully understand the complex relationships between various lifestyle factors and mental health outcomes among university students. Existing studies often focus on single aspects of lifestyle in isolation, overlooking the potential interactions between different factors. Moreover, the diversity of student populations in terms of age, gender, socioeconomic background, and cultural context suggests that these relationships may vary widely across different groups.

Thus, the objective of this study is to describe the lifestyle and levels of mental health of higher education students and determine their predictors and intercorrelations. Focusing on the factors most highlighted in the literature, the following research questions are outlined:(1)What lifestyle variables have the greatest predictive weight on the mental health of university students?(2)Do academic stress levels, time management, sleep patterns, diet, and substance use influence symptoms of anxiety and depression?(3)Is there an interaction between different lifestyle factors and mental health levels (e.g., the relationship between sleep, diet, and stress)?

## 2. Materials and Methods

A quantitative, cross-sectional, and descriptive study was conducted from January to May 2024 using an online questionnaire available to participants on the Google^®^ Forms platform.

### 2.1. Participants

A non-probabilistic convenience sampling method, which makes it easier to provide cost-effective access to the population in question, was chosen. Recruitment took place via written invitation disseminated through the Polytechnic of Porto students’ institutional emails. To be eligible, participants had to meet the following inclusion criteria: age of 18 or older, currently a student at one of the Polytechnic of Porto schools, and proficiency in Portuguese. The sample size was calculated for a margin of error of 5% (because it balances precision and practicality) and a reliability of 95% (because it minimizes Type I errors while maintaining reasonable certainty). These statistical standards are widely accepted in research and policy-making as standard thresholds ([27]). From a population of 21,000 Polytechnic of Porto students, a total of 378 participants were required according to the [65] ([65]). A statistical power analysis was conducted to confirm that 745 students were sufficient to detect the expected effects, and a value of 0.95 was obtained, confirming its adequacy ([75]).

### 2.2. Instruments

A sociodemographic questionnaire was used, covering age, gender, education area (health, education, engineering, arts, marketing, and media and design), education level (associated degree—two-year degree between a high school diploma and a bachelor’s degree, bachelor’s degree, master’s degree, and PhD degree), curricular year, working student (Y/N), living away from home (Y/N), weight and height, and self-perceived health status (range from 1 to 10). The FANTASTICO Lifestyle Questionnaire, which is used to assess wellbeing and lifestyle choices, and the DASS-21 and CORE-18, which are used to assess mental status, were selected.

The FANTASTICO Lifestyle Questionnaire ([42]; [72]) is a self-report instrument that explores habits and behaviors to evaluate the population’s lifestyles. It was chosen due to its comprehensive nature in assessing multiple lifestyle domains and its applicability to student populations. The questionnaire includes a total of 30 questions (items), all of which are multiple choice and explore ten domains of the physical, psychological, and social components of lifestyle, represented by the acronym “FANTASTICO”, where F—Family and Friends; A—Physical Activity/Associations; N—Nutrition; T—Tobacco; A—Alcohol and Other Drugs; S—Sleep/Stress; T—Work/Personality Type; I—Introspection; C—Health and Sexual Behaviors; and O—Other Behaviors. The items have three response options represented by a numerical value of 0, 1, or 2. The alternatives are arranged in three lines to facilitate coding, with the first alternative (1st line) always representing the highest value or the closest relation to a healthy lifestyle. The coding of the questions is performed as follows: 2 points for the 1st line, 1 point for the 2nd line, and 0 points for the 3rd line. By adding these values within each domain and multiplying them by two, the corresponding score for each domain is obtained. The sum of all points across all domains yields a global score that classifies individuals from 0 to 120 points. The “Guide for Healthy Universities and Other Higher Education Institutions” ([42]) proposes five classification levels that stratify behavior as follows: 0 to 46 (Needs Improvement); 47 to 72 (Fair); 73 to 84 (Good); 85 to 102 (Very Good); and 103 to 120 (Excellent). The lower the score, the greater the need for behavioral change. Generally, the results can be interpreted as follows: “Excellent” indicates that the lifestyle has a highly positive influence on health; “Very Good” indicates that the lifestyle has a sufficient positive influence on health; “Good” indicates that the lifestyle brings many health benefits; “Fair” means that the lifestyle provides some health benefits but also carries some risks; and “Needs Improvement” indicates that the lifestyle presents many risk factors. The Portuguese version of this instrument obtained a Cronbach’s α of 0.71 ([72]). In this study, the Cronbach’s α value is 0.76.

The Depression Anxiety Stress Scales (DASS-21) is a set of three self-report scales designed to measure the emotional states of depression, anxiety, and stress ([50]; [62]). It is a widely used and validated tool that could provide a reliable assessment of psychological distress, which is essential for understanding the mental health challenges faced by higher education students. These scales are designed to measure psychological distress dimensionally rather than categorically, focusing on the severity of symptoms rather than specific diagnostic thresholds. The total score from the DASS-21 offers a general assessment of psychological distress ([33]). The three scales are composed of seven items each, totalizing 21 items. Each item consists of a sentence or statement referring to negative emotional symptoms. Participants are asked to respond whether the statement applied to them “in the past week”. For each statement, there are four response options presented on a Likert scale. Participants rate the extent to which they experienced each symptom during the past week on a 4-point scale as follows: “0—did not apply to me at all”; “1—applied to me sometimes”; “2—applied to me often”; and “3—applied to me most of the time”. The DASS-21 is intended for individuals over 17 years old. The results of each scale are obtained by summing the scores of the corresponding seven items. The scale provides three scores, one for each subscale, where the minimum is “0” and the maximum is “21”. Higher scores on each scale indicate more negative affective states. The items of the 21-item DASS were selected so that they can be converted into scores of the full 42-item scale by multiplying the score by two. The Cronbach’s α values of the Portuguese version are 0.85 for the depression scale, 0.74 for the anxiety scale, and 0.81 for the stress scale ([62]). In this study, the Cronbach’s α values are 0.91 for the depression scale, 0.87 for the anxiety scale, and 0.90 for the stress scale.

Clinical Outcomes in Routine Evaluation (CORE)-18 is a self-report measure of psychological state, which, due to its broad-spectrum nature, allows for a wide variety of problems associated with mental health difficulties to be captured in addition to typical symptom measures ([22]). It offers insights into the emotional and mental health statuses of students beyond diagnostic categories, allowing for a more holistic evaluation of well-being. The participant answers 18 questions about how they have been feeling in the past week using a 5-point scale. The scale covers four dimensions: subjective well-being, problems/symptoms, life functioning, and risk/impairment. CORE-18 has two forms (A and B; A was used), derived from the original 36-item CORE-OM, conceived as very comparable but not identical measures and is thus suited for weekly clinical outcome monitoring due to the minimization of memory effects. Both short forms include all four of the subjective well-being items from the complete CORE-OM, so the well-being domain scores are strictly comparable across versions. Each item is rated on a scale ranging from “Never” to “Always or Almost Always”, with values ranging from 0 to 4 points. The overall score is obtained by adding the responses to the 18 items, generating a total score between 0 and 72 ([8]). The Cronbach’s α values were 0.90 both for the original and Portuguese versions of the CORE-OM ([22]; [68]). In this study, the Cronbach’s α value is 0.92.

### 2.3. Procedures

Ethical approval for this study was sought and received from the Ethics Committee of the School of Health (No. CE0087D). Before completing the online questionnaires, students were asked to give their informed consent.

The questionnaire was distributed via the institutional emails of students in the various schools of the Polytechnic of Porto in January 2024, beginning with an explanation of the study and the collection of informed consent. The scales were presented consecutively, taking approximately 10 min to complete. The data were transferred to an Excel spreadsheet and converted into numerical values. The statistical analysis was carried out using the software IBM^®^ SPSS^®^ version 29 for Windows with a significance level (α) of 0.05 for all statistical tests used. Descriptive statistics were used to characterize the sample, namely the mean (x) and standard deviation (sd) for continuous or discrete variables and frequencies (N; %) for nominal or ordinal data. The normality of variables was assessed through the Shapiro–Wilk test or the examination of data distribution using threshold criteria for skewness and kurtosis, aiming for values less than |2.0| and |9.0|, respectively ([29]). Comparisons between the assessment instruments (total score and subscales) and the variables were made using independent Student’s *t*-tests and one-way ANOVA. Pearson’s correlation coefficients were used to assess the association between the instrument values and age, body mass index (BMI), and perceived health status ([53]).

Multiple Linear Regression models using a stepwise method were implemented to predict the value of dependent variables (total CORE-18, total DASS-21, and total FANTASTICO). Using stepwise methods for variable selection in multiple regression can save time and effort, particularly with many potential predictors. Categorical variables were added to the model as dummy variables. The assumptions for the analysis were tested and verified—namely, a linear relationship between the independent and dependent variables was determined using a scatter plot; the absence of multicollinearity was determined using values of tolerance such as the Variance Inflation Factor (VIF); the independence of the residuals was determined using the Durbin–Watson test; constant variance was determined using a graph of “standardized residuals” against the “standardized predicted value”; a normal distribution of residuals was determined using a quantile–quantile (Q-Q) plot; and the presence of outliers was determined using Cook’s distance values less than 1 ([53]). Potential confounding variables, such as socioeconomic status or prior mental health history, which could influence the results, were included in linear regression models. However, none of them showed any influence on the response variable, so the best models for each of the regressions performed were those reported, and none of them included potentially confounding variables.

## 3. Results

A total of 756 students answered the online questionnaire, but only 745 were considered (Table 1) due to incomplete filling of the questionnaire. The mean age of this sample was 23.22 (±6.82) years, most of them were women (69.90%), and 8.00% described themselves as non-binary. The majority were bachelor students (87.00%) from health (41.50%), education (14.20%), engineering (25.40%), and other educational areas (19.90%), and most of them were first-year (38.00%) or third-year (30.50%) students and living with family (74.00%). Worker students represented 31.00% of the sample. The participants had an average BMI of 23.18 (±4.08), indicating a healthy weight, and perceived their state of health as 7.08 (±1.46) on a scale from 1 to 10.

The findings and their implications should be discussed in the broadest context possible. Future research directions may also be highlighted. Table 2 shows the values of perceived lifestyle and mental health, measured by the FANTASTICO (84.73 (±12.86)), CORE-18 (23.49 (±9.82)), and DASS-21 subscales (depression—12.54 ± 10.68; anxiety—9.38 ± 8.30; stress—17.25 ± 10.35) for the total sample. By analyzing these values according to the sociodemographic variables, also shown in Table 2, it is visible that statistically significant differences exist between genders (CORE-18 F(2) = 16.14, *p* < 0.001; DASS-21 Depression F(2) = 4.49, *p* = 0.012; DASS-21 Anxiety F(2) = 21.45, *p* < 0.002; DASS-21 Stress F(2) = 41.52, *p* < 0.001; FANTASTICO F(2) = 3.37, *p* = 0.035). Post hoc tests (Bonferroni) show differences between females and males in the total CORE-18 scores (5.34 ± 1.01; *p* < 0.001), in the DASS-21 anxiety subscale (3.79 ± 0.65; *p* < 0.001), and in the DASS-21 stress subscale (7.04 ± 0.791; *p* < 0.001) and between males and non-binary individuals (8.91 ± 3.34; *p* < 0.001). In all of these cases, the differences found are favorable to male participants’ mental health.

The results in the DASS-21 depression (F(3) = 2.65, *p* = 0.048) and stress (F(3) = 3.13, *p* = 0.025) subscales and in FANTASTICO (F(3) = 7.07, *p* < 0.001) also show differences according to the BMI levels. Post hoc tests (Bonferroni) show differences in FANTASTICO between individuals with a normal weight and with pre obesity (3.27 ± 1.23; *p* = 0.047) and between individuals with a normal weight and obesity (8.04 ± 2.05; *p* < 0.001), with both cases favoring participants with a normal weight, and in the DASS-21 stress subscale, there are differences between individuals classified as underweight and those with pre obesity (4.98 ± 1.73; *p* = 0.024), with higher stress being found among the former.

Concerning education area, differences were found in the DASS-21 depression subscale (F(3) = 2.80, *p* = 0.040), the DASS-21 stress subscale (F(3) = 5.47, *p* = 0.001), and FANTASTICO (F(3) = 4.88, *p* = 0.002). Post hoc tests (Bonferroni) show higher levels in the total FANTASTICO scores among health students compared to those from the “others” category (4.65 ± 1.30; *p* = 0.002). The DASS-21 stress subscale is higher in health students than among engineering students (3.52 ± 0.947; *p* = 0.001). Bachelor’s students’ FANTASTICO scores are superior to those of master’s and PhD students (4.23 ± 1.51; *p* = 0.015).

Being a student living away from the family resulted in worse scores in CORE-18 (t(733) = 2.70, *p* = 0.007), the DASS-21 depression (t(743) = 2.25, *p* = 0.025) and anxiety subscales (F(743) = 2.24, *p* = 0.026), and lifestyle, as measured by FANTASTICO (t(743) = −3.21, *p* = 0.001).

An association between the DASS-21 subscales and CORE-18 total and subscales was found (Table 3), with Pearson’s r = 0.591 (*p* < 0.001) in the correlation of the DASS-21 stress subscale and the CORE-18 subjective well-being subscale, and between the CORE-18 total and CORE problem symptoms subscale (Pearson’s r = 0.956; *p* < 0.001). A negative association of the DASS-21 subscales and CORE-18 total and subscales was found with FANTASTICO, with Pearson’s r = −0.467 (*p* < 0.001) in the correlation with the DASS-21 anxiety subscale and Pearson’s r = −0.641 (*p* < 0.001) in the correlation with the total CORE-18 score. Perceived health status was also negatively associated with the DASS-21 subscales and CORE-18 total and subscales (Pearson’s r = −0.393, *p* < 0.001 in DASS-21 anxiety subscale and Pearson’s r = −0.478, *p* < 0.001 in total CORE-18 score) and positively associated with FANTASTICO (Pearson’s r = 0.444; *p* < 0.001).

We also found a negative association between the FANTASTICO subscales, except for the Other Behaviors subscale, and the CORE-18 subscales, with Pearson’s r = −0.702 (*p* < 0.001) in the correlation of the FANTASTICO Introspection subscale and the CORE-18 functioning difficulties subscale and Pearson’s r = −0.080 (*p* = 0.030) between the FANTÁSTICO Alcohol and Other Drugs subscale and the CORE-18 subjective well-being deficits subscale. A negative association between the DASS-21 subscales and FANTASTICO subscales was also found, with Pearson’s r = −0.685 (*p* < 0.001) in the correlation between the FANTASTICO Introspection subscale and the DASS-21 depression subscale, and Pearson’s r = −0.091 (*p* = 0.013) in the correlation between the FANTASTICO Health and Sexual Behavior subscale and the DASS-21 anxiety subscale. Perceived health status was also associated with the FANTASTICO subscales, with Pearson’s r = 0.451 and *p* < 0.001 in the FANTASTICO Sleep and Stress subscale and Pearson’s r = 0.081 and *p* = 0.027 in the FANTASTICO Health and Sexual Behavior subscale.

Multivariate linear regression (Table 4) revealed statistically significant models for FANTASTICO [F(5,718) = 142.35; *p* < 0.001; R2 = 0.50], CORE [F(6,724) = 336.13; *p* < 0.001; R2 = 0.71], and DASS [F(5,718) = 346.37; *p* < 0.001; R2 = 0.72]. The DASS-21 depression subscale is a predictor of both CORE-18, where a higher level of depression symptoms was associated with an increase of 0.46 in the total score of CORE-18 (B = 0.46; t = 11.76; *p* < 0.001), and FANTASTICO, where a better lifestyle was associated with as increase of 4.18 in the total score of CORE-18 (B = 4.18; t = −9.01; *p* < 0.001). The DASS-21 anxiety (B = 0.18; t = 8.84; *p* < 0.001) and DASS-21 stress subscales (B = 0.28; t = 6.43; *p* < 0.001) were also found to be predictors of CORE-18, with the model explaining 74.00% of the results. Other predictors of CORE-18 were the FANTASTICO score (B = −0.21; t = −8.29; *p* < 0.001), a worse perceived health state, which was associated with a decrease of 0.54 in the total score of CORE-18 (B = −0.54; t = −2.78; *p* = 0.006), and the male sex, which was linked to a decrease of 1.63 in the total score of CORE-18 (B = −1.63; t = −2.73; *p* = 0.006). The male sex was linked to a decrease of 4.33 in the total score of DASS-21 (B = −4.33; t = −3.54; *p* < 0.001), and a worse perceived health state was also associated with a decrease of 1.72 in the total score of DASS-21 (B = −1.72; t = −4.07; *p* < 0.001). Other predictors of the total score of DASS-21 included the total score of CORE-18, where having more mental health difficulties was associated with an increase of 1.62 in DASS-21 (B = 1.62; t = 33.33; *p* < 0.001), being a working student was associated with an increase of 2.57 in DASS-21 (B = 2.57; t = 2.18; *p* = 0.030), and being a student of a program in the education field was associated with a decrease of 3.27 in the total score of DASS-21 (B = −3.27; t = −2.07; *p* = 0.039). This model explained 71.00% of the results. Predictors of a better lifestyle (in a model that explained 50.00% of the results) included, in addition to the DASS-21 depression subscale, the total score of CORE-18, where having more mental health difficulties was associated with a decrease of 0.42 in lifestyle (B = −0.42; t = −9.00; *p* < 0.001); perceived health status, where a better perceived health state was associated with an increase of 1.31 in lifestyle (B = 1.31; t = 4.82; *p* < 0.001); being a woman, which was linked to an increase of 5.20 in lifestyle (B = 5.20; t = 6.83; *p* = 0.010); and being underweight, which was also linked to an increase of 2.01 in lifestyle (B = 2.01; t = 2.67; *p* = 0.008).

## 4. Discussion

The objective of this study was to describe the lifestyle and levels of mental health of higher education students and explore their predictors. The results indicate that Polytechnic of Porto students generally exhibit a lifestyle close to very good. However, they also experience mild levels of depression, anxiety, and stress that are close to the threshold for moderate according to established cut-off points, and they appear to stand slightly above the clinical cut-off for psychological distress.

Our analysis revealed several significant predictors of mental health and lifestyle outcomes among students. The DASS-21 depression subscale emerged as a significant predictor for both the CORE-18 and FANTASTICO scores. This finding aligns with existing research that highlights the strong relationship between depression and overall well-being. For instance, prior studies have consistently shown that higher levels of depression are associated with poorer mental health outcomes and lower quality of life ([23]). Our study also identified that anxiety and stress, as measured by DASS-21, are predictors of CORE-18 scores. Recent studies have explored the relationship between stress and decreased well-being in students’ mental health, emphasizing significant findings in this area. Stress is strongly linked to decreased psychological well-being among students, often leading to negative outcomes such as anxiety, depression, and lower life satisfaction ([57]). For instance, research shows that higher perceived stress levels are associated with poorer cognitive functioning and lower academic performance among students ([3]). Additionally, resilience has been identified as a critical moderator where students with higher resilience tend to maintain better well-being despite experiencing stress ([44]; [61]). Moreover, perceived health status was negatively associated with the DASS-21 subscales and CORE-18 scores, emphasizing the importance of self-perceived health in mental well-being and reinforcing the need for comprehensive health assessments in predicting mental well-being ([14]). Studies have reported that perceived health was more strongly correlated with subjective well-being than objectively measured health ([71]; [47]).

This finding is consistent with research indicating that poor self-rated health is a strong predictor of mental health issues ([37]). A 2023 Portuguese study ([76]) found that among 1447 university students surveyed, high levels of anxiety (66.7%), depression risk (37.3%), and low resilience (24.9%) were prevalent, with the male gender, study time, employment, extracurriculars, and physical exercise being linked to better mental health outcomes, while excessive news consumption, online lecture difficulties, and social isolation were associated with worse psychological indicators.

Our study also found that perceived health status and being men were significant negative predictors of both the total scores for the DASS-21 scales and the CORE-18. This supports previous research suggesting that gender differences can influence mental health outcomes, with men often experiencing different stressors and coping mechanisms compared to women ([54]). In fact, in our study, women scored worse on DASS-21 and CORE-18 compared to men. This disparity may be attributed, in addition to the already noted factors such as higher stress, anxiety, and emotional reactivity ([41]), to differences in coping mechanisms, as suggested by [54] ([54]), where women tend to engage in emotion-focused coping strategies more frequently than men. For instance, a 2024 study found that female students reported higher levels of anxiety and depressive symptoms, which could be linked to greater stress and emotional reactivity, as well as a tendency to use emotion-focused coping strategies more frequently than men. These findings align with another study that highlighted gender-specific differences in how stress is perceived and managed, with women showing a higher prevalence of mental health issues under similar stress conditions ([74]).

Additionally, a comprehensive survey from the Healthy Minds Network revealed that female students are more likely to seek mental health services and report higher levels of psychological distress than male students. This may be since women are generally more open to acknowledging their mental health issues, while men may be more likely to suppress or externalize these problems ([32]). Furthermore, traditional gender roles often impose greater stress on women, who are typically expected to balance work, family, and social obligations, further exacerbating their mental health challenges ([55]).

Our study also shows a strong positive correlation between CORE-18 scores and DASS-21 scores, indicating that as psychological suffering increases, so do depression, anxiety, and stress levels. Recent studies emphasize a significant correlation between higher levels of depression and increased stress and anxiety among university students. This relationship is particularly pronounced due to the demanding academic environment, financial pressures, and social challenges faced by students. For instance, a 2023 scoping review on health science students found that depression and anxiety were prevalent due to stress factors such as heavy workloads and personal issues. The review also noted that these mental health issues often co-occur, exacerbating one another and leading to a cycle of increasing psychological distress ([1]). Another study observed that high levels of depression among higher education students are strongly associated with increased stress and anxiety, contributing to a decline in overall mental health ([20]).

Moreover, this psychological distress often correlates with a poorer perceived health status, as mental health significantly impacts how individuals perceive their overall well-being and as people engage in fewer health-promoting behaviors due to their psychological state ([19]), which appears to be in line with our findings concerning the associations between mental health, lifestyle, and perceived health status.

Likewise, the correlation found between higher levels of depression and increased stress and anxiety is common. Depression, stress, and anxiety are often interrelated—high levels of stress can trigger or exacerbate depressive symptoms, and individuals with depression are more likely to experience heightened anxiety ([56]). These conditions feed into each other, worsening the overall mental health of the individual—when one aspect of mental health deteriorates, it often negatively impacts other areas, leading to compounded psychological distress. Students experiencing higher levels of stress, anxiety, and depression reported significantly greater psychological distress, impacting their ability to function effectively in daily life and increasing their risk for long-term mental health issues. The stressors related to academic pressure, social isolation, and uncertainty about the future have exacerbated these issues, particularly in the post-pandemic context ([38]; [82]). 

Our findings suggest that being a woman and having an underweight BMI were predictors of a better lifestyle measured by the FANTASTICO Lifestyle Questionnaire. Several studies have reported an association between BMI and lifestyles, indicating that a worse lifestyle has a negative impact on BMI as it leads to overweight and obesity ([83]; [40]; [73]). However, studies showing the influence of BMI on lifestyle are scarce. [91] ([91]) reported that losing weight was more likely to be reflected in better quality of life in healthy people, and [43] ([43]) reported the highest quality of life among women with a normal BMI. Our study shows that being underweight is predictive factor of a better lifestyle, and this could also be a contribution to this topic.

Concerning gender differences, women exhibit a healthier lifestyle compared to men across several dimensions, including “family and friends”, “nutrition”, “tobacco”, “addiction and driving”, “work and personality”, “health and sexual behaviors”, and “other behaviors”, as well as in the overall health behavior score. These findings are consistent with the existing literature, which indicates that women generally adopt healthier lifestyles ([58]), including better dietary habits, fewer risk behaviors, and lower consumption of tobacco and alcohol ([90]; [34]; [59]). Similarly, research on university students in Colombia identified that men are more likely to engage in risky behaviors, including higher rates of tobacco and alcohol consumption, while women tended to have better health practices like walking daily ([70]).

Studies have shown that women are often more health-conscious and more likely to seek medical care, but this can also mean they are more aware of and report more health issues, which could lower their perceived lifestyle quality ([77]). These outcomes may be attributed to greater dissatisfaction with body weight and the significance of body image for women ([9]), or to a heightened perception of risk situations ([45]). Conversely, women exhibit a less healthy lifestyle in dimensions such as “physical activity”, “sleep and stress”, and “introspection” in the present study. Gender differences in lifestyle choices have been well-documented across various studies, highlighting significant disparities in behaviors related to health and well-being, with women reporting more significant challenges with emotional and mental health ([70]). For example, a study conducted among senior high school students in South Korea found that girls exhibited more unhealthy lifestyle behaviors, such as lower physical activity and poorer sleep satisfaction, and were more vulnerable to mental health issues like stress and depression. In contrast, boys were more prone to risky behaviors such as smoking and drinking ([41]). Also, the study by [4] ([4]), which found that women experience more sleep disturbances compared to men, and the studies by [9] ([9]), [67] ([67]), and [11] ([11]), which reported higher levels of physical activity among men, confirm these data.

Accordingly, a study in Australia with 6949 students showed that men engage more in physical activity but have a less adequate diet, exactly contrary to women ([49]). The lower levels of physical activity among women may be attributed to discomfort during exercise or a lack of motivation, energy, and time ([21]; [67]). These lower levels of physical activity seem to negatively affect sleep quality ([46]), which could explain the lower scores in this dimension in our study. Regarding sleep disturbances, research has consistently shown that they are prevalent among university students, with female students generally reporting poorer sleep quality compared to their male counterparts. Factors contributing to this disparity include higher levels of stress, anxiety, and emotional reactivity among women ([41]). For instance, a study conducted in Jiangsu Province, China, found that being a woman was associated with a significantly higher likelihood of poor sleep quality. This research highlighted that female students were 23.8% more likely to experience sleep disturbances compared to male students, which could be linked to higher stress levels and emotional concerns prevalent among women ([36]).

Despite engaging in healthier lifestyle behaviors, women often report worse mental health outcomes than men, a disparity that may be influenced by societal and cultural factors. Women are more likely to experience gender-based violence, low self-esteem, and the pressures associated with traditional caregiving roles, all of which contribute to increased psychological distress. Additionally, societal expectations often lead women to internalize stress, resulting in higher rates of mood and anxiety disorders. Conversely, men may underreport mental health issues due to cultural norms that discourage expressing vulnerability, leading to an underestimation of their psychological distress ([60]).

In clinical populations, gender differences in lifestyle factors and health outcomes have been observed, though the patterns may differ from those in the general population. For instance, among individuals with diabetes mellitus, studies indicate that men are more likely to engage in unhealthy behaviors such as poor diet, smoking, alcohol consumption, and sedentary lifestyles. Conversely, women with diabetes tend to experience higher rates of depression, obesity, and insufficient physical activity ([83]). Additionally, research has shown that women with diabetes have a 58% greater risk of coronary heart disease and a 13% higher risk of all-cause mortality compared to men with the same condition ([84]). These findings suggest that while some gender differences in lifestyle factors are consistent across populations, individuals with health conditions like diabetes may exhibit unique patterns that warrant targeted interventions.

The findings of this study highlight a significant negative correlation between some of the subscales of the FANTASTICO questionnaire—namely, “Family and Friends”, “Sleep/Stress”, “Work/Personality Type”, and “Introspection”—and the DASS-21 and CORE-18 scales. These results underscore the protective role that supportive social relationships, an optimist and fulfilled stance, reflective practices, stress management, and healthy sleep patterns play in mitigating distress levels among higher education students. These findings align with the current literature, which highlights the importance of social support networks ([69]), optimism and life satisfaction ([15]), mindfulness and self-reflection ([28]), and adequate sleep ([2]) in fostering mental well-being. Future interventions aimed at improving students’ mental health should consider promoting these lifestyle factors as part of comprehensive support strategies. In our study, individuals with a bachelor’s degree tend to have better lifestyle scores compared to those with master’s or doctoral degrees. Bachelor’s degree holders are often in positions that demand less time and energy compared to roles requiring advanced degrees. Individuals with master’s or doctoral degrees are more likely to occupy high-pressure roles that involve long hours, intense focus, and significant responsibilities, leaving them with less time and energy to maintain a healthy lifestyle ([39]). Conversely, those with associate degrees may also exhibit poorer lifestyle habits, possibly due to dissatisfaction with their current situation and a desire to pursue a bachelor’s degree.

Living away from home was associated with worse scores for CORE-18, FANTASTICO, and the DASS-21 anxiety and depression subscales. Living independently often leads students to make worse lifestyle choices since they frequently experience social isolation, losing their familiar support networks and facing loneliness in unfamiliar environments. This isolation, coupled with economic instability, limits access to essential resources like healthcare and nutritious food. The challenges of adapting to a new culture and environment may negatively impact their overall lifestyle and well-being as well ([88]).

Balancing the demands of both work and study can be exceptionally challenging. Working students must navigate the dual responsibilities of attending classes, completing assignments, and preparing for exams, all while fulfilling their work obligations. This continuous pressure to manage time effectively can intensify anxiety, particularly when deadlines for both work and academic tasks coincide ([66]). At first sight, our study did not yield statistically significant differences in lifestyle and mental health measures between students who solely focus on their studies and those who also work, possibly due to the varying number of hours worked (part-time vs. full-time). Still, being a working student was a significant positive predictor of the global depression, anxiety, and stress scores, supporting the vulnerability of this population to mental health issues.

Overall, evidence supports the positive impact of healthy lifestyle choices on academic performance and mental well-being. Students who, for example, adopt healthier eating habits and engage in regular physical activity tend to experience lower levels of distress and perform better academically ([35]; [31]). This underscores the importance of integrating lifestyle interventions into academic settings. Given the persistent and widespread nature of mental illness among higher education students ([51]), there is a pressing need for systemic changes that address these issues comprehensively. Our results suggest that targeted interventions focusing on mental health and well-being are crucial for supporting students, particularly in education settings, such as lifestyle redesign programs ([6]). Establishing mental health and well-being groups within academic institutions could provide students with a supportive network where they can build trusting relationships. Such groups might increase students’ willingness to share important information and seek help before reaching a crisis point, ultimately promoting a more proactive approach to mental health management. This aligns with Sustainable Development Goal 3, Good Health and Well-being ([81]), which emphasizes the necessity of ensuring healthy lives and enhancing well-being through comprehensive healthcare. By incorporating preventive measures and mental health support within the academic environment, universities can contribute significantly to achieving this global goal.

Despite the valuable insights provided by this study, there are some limitations that must be acknowledged. First, the sample was limited to a single higher education institution, which may restrict the generalizability of the findings. Future studies should consider more diverse samples and include other variables that may influence mental health, such as social support and economic pressures. Second, while FANTASTICO provides valuable insights into students’ lifestyle choices, it may not fully capture the broader aspects of quality of life. Incorporating an additional instrument specifically designed to evaluate overall quality of life (also including dimensions such as physical health, psychological well-being, and social relationships) could offer a more comprehensive understanding of students’ well-being and provide a more holistic view of how lifestyle factors impact students’ overall health. Finally, the cross-sectional design of this study limits the ability to establish causal relationships between lifestyle factors and mental health outcomes as it captures associations at a single point in time rather than over a longitudinal trajectory. Additionally, the reliance on self-report measures may introduce common method bias, potentially inflating correlations due to shared variance ([64]). Future studies could mitigate this bias by integrating objective assessments, such as wearable activity trackers for physical activity and sleep or clinical evaluations for mental health conditions, to enhance the robustness of the findings ([30]).

## 5. Conclusions

This study provides a comprehensive analysis of the mental health and lifestyle factors among higher education students at the Polytechnic of Porto, revealing significant predictors that influence their well-being. While students generally exhibited a favorable lifestyle, they also experienced mild levels of depression, anxiety, and stress, which approach moderate thresholds.

Our findings emphasize the critical role of mental health in shaping overall lifestyle outcomes, with the DASS-21 Depression subscale being a strong predictor for both lifestyle and mental health measures. The significant correlations observed between depression, anxiety, stress, and lifestyle highlight the interrelated nature of these factors, underscoring the need for integrated approaches in addressing student well-being. Gender differences were also evident, with women reporting worse mental health outcomes but engaging in healthier lifestyle behaviors compared to men. Additionally, factors such as perceived health status, education level, and living way from home were found to significantly impact both lifestyle and mental health, further emphasizing the complex interplay of personal, social, and academic factors in students’ lives.

These results suggest implementing mental health support systems and promoting healthier lifestyle choices within academic institutions to ultimately enhance students’ academic success and overall quality of life. Future research should focus on more diverse samples and consider additional variables such as social support and economic pressures to deepen the understanding of student well-being. Moreover, adopting longitudinal designs would help clarify the causal relationships between lifestyle factors and mental health, ultimately guiding more effective interventions.

## Figures and Tables

**Table 1 behavsci-15-00253-t001:** Sociodemographic characterization of the sample.

Variables	n (%)
Gender	
Men	218 (29.30)
Woman	521 (69.90)
Non-binary	6 (0.08)
Education Area	
Health	309 (41.50)
Education	106 (14.20)
Engineering	189 (25.40)
Others	149 (19.90)
Education Level	
Associated degree	16 (2.10)
Bachelor’s degree	648 (87.00)
Master’s and PhD degrees	81 (10.90)
Curricular Year	
First year	283 (38.00)
Second year	169 (22.70)
Third year	227 (30.50)
Fourth year	66 (8.90)
Student Worker	
Yes	231 (31.00)
No	514 (69.00)
Living away from family home	
Yes	194 (26.00)
No	551 (74.00)
**Variables**	**M (SD)**
Age (years)	23.22 (6.82)
BMI	23.18 (4.08)
Self-perceived health status	7.08 (1.46)

M—Mean; SD—Standard Deviation; BMI—Body Mass Index.

**Table 2 behavsci-15-00253-t002:** Lifestyle and students’ mental health measures according to sociodemographic variables.

Variables	CORE-18 Total		DASS-21 Depression		DASS-21 Anxiety		DASS-21 Stress		FANTASTICO Total	
	M(*SD*)	*p*	M(*SD*)	*p*	M(*SD*)	*p*	M(*SD*)	*p*	M(*SD*)	*p*
Total sample	23.49 (12.72)		12.54 (10.68)		9.38 (8.30)		17.25 (10.35)		84.73 (12.86)	
Gender Male Female Non-binary	19.63 (12.39) 24.98 (12.48) 34.17 (14.47)	<0.001	10.92 (10.14) 13.14 (10.84) 19.00 (9.53)	0.012	6.63 (7.46) 10.42 (8.32) 19.33 (8.45)	<0.001	12.22 (9.58) 19.26 (9.90) 25.33 (11.84)	<0.001	83.44 (13.84) 85.37 (12.41) 75.33 (7.23)	0.035
Education Area Health Education Engineering Others	23.28 (12.99) 23.81 (11.45) 22.24 (12.79) 25.36 (12.82)	0.178	12.28 (10.94) 11.40 (10.28) 11.92 (10.32) 14.79 (10.67)	0.040	9.64 (8.52) 9.08 (7.59) 8.54 (8.44) 10.18 (8.13)	0.296	18.24 (10.50) 17.25 (9.35) 14.72 (11.06) 18.48 (9.25)	0.001	86.54 (18.09) 85.13 (19.68) 83.66 (18.51) 81.89 (26.99)	0.015
Education Associated degree Bachelor’s degree Master’s and PhD degrees	26.69 (15.22) 23.09 (12.76) 26.00 (11.54)	0.093	16.38 (11.64) 12.38 (10.66) 13.11 (10.60)	0.294	11.00 (10.40) 9.23 (8.21) 10.25 (8.60)	0.430	17.50 (11.88) 16.95 (10.36) 19.63 (9.74)	0.089	78.36 (13.86) 85.34 (12.68) 81.11 (13.30)	0.003
Curricular year First year Second year Third year Fourth year	24.12 (12.87) 24.57 (13.11) 22.79 (11.87) 20.43 (13.51)	0.097	13.38 (10.49) 12.80 (11.11) 11.92 (10.47) 10.39 (10.88)	0.153	9.72 (8.36) 10.17 (8.59) 8.74 (7.81) 8.12 (8.80)	0.182	17.39 (10.45) 17.83 (10.74) 17.22 (9.78) 15.30 (10.78)	0.404	84.25 (13.23) 83.43 (12.27) 85.70 (12.05) 86.73 (15.06)	0.172
BMI Underweight Normal weight Pre obesity Obesity	26.28 (13.80) 23.18(12.77) 23.33(12.20) 24.98 (13.10)	0.379	16.29 (12.31) 12.18 (10.51) 12.16 (10.71) 14.38 (10.70)	0.048	10.83 (9.42) 9.28 (8.19) 8.92 (8.49) 10.09 (7.53)	0.517	20.75 (9.77) 17.19 (10.56) 15.77 (9.91) 18.81 (8.50)	0.025	82.86 (13.31) 85.94 (12.80) 82.67 (12.66) 77.90 (11.50)	<0.001
Student worker Yes No	24.45 (13.23) 23.06 (12.47)	0.172	12.45 (10.22) 12.58 (10.89)	0.878	8.90 (8.26) 9.60 (8.32)	0.288	17.67 (10.44) 17.06 (10.31)	0.458	83.69 (13.59) 85.19 (12.50)	0.139
Living away from family home Yes No	25.60 (13.06) 22.73 (12.52)	0.007	14.02 (11.27) 12.02 (10.43)	0.025	10.53 (8.87) 8.98 (8.06)	0.026	17.88 (11.05) 17.03 (10.09)	0.328	82.20 (12.94) 85.62 (12.72)	<0.001

M—Mean; SD—Standard Deviation; BMI—Body Mass Index; CORE-18—Clinical Outcomes in Routine Evaluation; DASS-21—Depression, Anxiety, and Stress Scales; FANTASTICO—FANTASTICO Lifestyle Questionnaire.

**Table 3 behavsci-15-00253-t003:** Association between mental health and lifestyle measures and sociodemographic variables.

	CORE-18_swe	CORE-18_ps	CORE-18_lfd	CORE-18_rh	DASS-21 Depression	DASS-21 Anxiety	DASS-21 Stress	FANTASTICO Total	PHS	Age	BMI
CORE-18 total	0.792 <0.001	0.956 <0.001	0.911 <0.001	0.853 <0.001	0.796 <0.001	0.689 <0.001	0.752 <0.001	−0.641 <0.001	−0.478 <0.001	−0.007 0.856	0.028 0.446
CORE-18_swe		0.683 <0.001	0.690 <0.001	0.527 <0.001	0.646 <0.001	0.483 <0.001	0.591 <0.001	−0.576 <0.001	−0.386 <0.001	0.013 0.715	0.052 0.156
CORE-18_ps			0.796 <0.001	0.805 <0.001	0.756 <0.001	0.647 <0.001	0.705 <0.001	−0.604 <0.001	−0.460 <0.001	0.017 0.651	0.040 0.275
CORE-18_lfd				0.714 <0.001	0.744 <0.001	0.640 <0.001	0.696 <0.001	−0.610 <0.001	−0.431 <0.001	−0.013 0.721	0.017 0.655
CORE-18_rh					0.650 <0.001	0.663 <0.001	0.666 <0.001	−0.469 <0.001	−0.405 <0.001	−0.056 0.129	−0.023 0.532
DASS-21 Depression						0.688 <0.001	0.743 <0.001	−0.618 <0.001	−0.442 <0.001	−0.078 0.033	0.031 0.407
DASS-21 Anxiety							0.760 <0.001	−0.467 <0.001	−0.393 <0.001	−0.062 0.093	0.007 0.854
DASS-21 Stress								−0.502 <0.001	−0.434 <0.001	−0.063 0.087	−0.009 0.801
FANTASTICO total									0.444 <0.001	−0.019 0.601	−0.145 <0.001
PHS										−0.126 <0.001	−0.165 <0.001
Age											0.243 <0.001

r—Pearson’s correlation; CORE-18 total—total clinical outcomes in routine evaluation; CORE-18_swe—core subjective well-being deficits subscale; CORE-18_ps—core problem symptoms subscale; CORE-18_lfd—core life functioning difficulties subscale; CORE-18_rh—core risk harm subscale; DASS-21—depression, anxiety and stress scales; FANTASTICO—FANTASTICO Lifestyle Questionnaire; PHS—perceived health status; BMI—body mass index.

**Table 4 behavsci-15-00253-t004:** Multiple regression models to analyze predictors of total scores of CORE-18, DASS-21, and FANTASTICO.

	Total CORE-18	Total DASS-21	FANTASTICO
Constant	33.48 (2.65)	13.41 (3.78)	87.71 (2.39)
DASS-21 Depression	0.46 ** (0.04)		−0.28 ** (0.05)
DASS-21 Anxiety	0.18 * (0.05)		
DASS-21 Stress	0.28 ** (0.04)		
FANTASTICO	−0.21 ** (0.03)		
Core-18 Total		1.62 ** (0.05)	−0.42 ** (0.05)
PHS	−0.54 ** (0.19)	−1.72 ** (0.42)	1.31 ** (0.27)
Working Student		2.58 * (1.18)	
Man	−1.63 ** (0.59)	−4.33 ** (1.22)	
Woman			5.20 ** (0.76)
BMI Underweight			2.01 ** (0.76)
Student Education Field		−3.27 * (1.59)	
R-Squared	0.74	0.71	0.50

Notes. Standard errors are reported in parentheses. PHS—perceived health status; BMI—body mass index; CORE-18 total—total clinical outcomes in routine evaluation; DASS-21—depression, anxiety, and stress scales, 21 items; FANTASTICO—FANTASTICO Lifestyle Questionnaire; * and ** indicate significance at the 95% and 99% levels, respectively.

## Data Availability

The data that support the findings of this study are available from one of the authors (M.J.T.) upon reasonable request.

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
