# Peer review of "Mental Health and Lifestyle Factors Among Higher Education Students: A Cross-Sectional Study"

_behavsci, 2025, doi:10.3390/bs15030253_

Round 1

Reviewer 1 Report

Comments and Suggestions for Authors

This is a very interesting paper investigating cross-sectionally the association between lifestyle factors and mental health levels in a sample of higher education students. 

The paper is well-written and of interest for the readers, however; several minor changes are recommended before considering the paper for publication. 

ABSTRACT.

The abbreviation of the DASS-21, CORE-18, and FANTASTICO Lifestyle Questionnaires should be previously described.

The results subsection of the abstract should be started with something like "We found...", to separate it from the methods subsection.

INTRODUCTION

The introduction section is mainly focused on the impact of physical activity, sleep quality, nutrition and social relationships, in the transition to university life. I recommend to add the role of the climate change /climate crisis on the association between mental health and life-style factors. It seems to be really important to work on a better lifestyle, to improve physical and mental health, in times of climate changes. Heat waves, increasing temperatures and air pollution are frequently affecting physical and mental health, as well as cognitive outcomes in people.

METHODS

The authors should briefly discuss why they have selected the FANTASTIVO Lifestyle questionnaire instead of other questionnaires. 

RESULTS

The authors report gender differences in mental health in association with lifestyle factors. Differences were favorable to male participants compared to women. I recommend to add a specific paragraph about gender differences.

DISCUSSION

Line 397. The authors discuss that women exhibited healthier lifestyle compared to men. These findings are based on the general population, but not in clinical populations (diabetes mellitus, people with mental health disorders). 

I recommend to add some words about clinical populations, to discuss if these gender differences apply to people with health problems, and are or not in the same line that findings from students' samples.

Author Response

This is a very interesting paper investigating cross-sectionally the association between lifestyle factors and mental health levels in a sample of higher education students. The paper is well-written and of interest for the readers, however; several minor changes are recommended before considering the paper for publication. 

Response: We would like to sincerely thank you for your time and effort in reviewing our manuscript. We appreciate your insightful comments and suggestions, which have helped us improve the quality and clarity of our work. Below, we provide a point-by-point response to your comments and outline the revisions made accordingly. All changes in the manuscript are highlighted in blue.

ABSTRACT.

The abbreviation of the DASS-21, CORE-18, and FANTASTICO Lifestyle Questionnaires should be previously described.

Response: We have now introduced the full names of these questionnaires before their abbreviations in the abstract to enhance clarity: Depression Anxiety Stress Scales (DASS-21); Clinical Outcomes in Routine Evaluation (CORE)-18; however, FANTASTICO description is quite extensive (“"FANTASTICO": F - Family and Friends; A - Physical Activity/Associations; N - Nutrition; T - Tobacco; A - Alcohol and Other Drugs; S - Sleep/Stress; T - Work/Personality Type; I - Introspection; C - Health and Sexual Behaviors; O - Other Behaviors”) and we believe saying FANTASTICO Lifestyle Questionnaire is enough.

The results subsection of the abstract should be started with something like "We found...", to separate it from the methods subsection.

Response: We have modified the beginning of the results section in the abstract to clearly distinguish it from the methods.

INTRODUCTION

The introduction section is mainly focused on the impact of physical activity, sleep quality, nutrition and social relationships, in the transition to university life. I recommend to add the role of the climate change /climate crisis on the association between mental health and life-style factors. It seems to be really important to work on a better lifestyle, to improve physical and mental health, in times of climate changes. Heat waves, increasing temperatures and air pollution are frequently affecting physical and mental health, as well as cognitive outcomes in people.

Response: We agree that climate change is an important factor affecting mental health. We have now included a paragraph on how climate change, including heat waves, increasing temperatures, and air pollution, impacts both physical and mental health, adding two references.

METHODS

The authors should briefly discuss why they have selected the FANTASTIVO Lifestyle questionnaire instead of other questionnaires. 

Response: We have now included a justification for the choice of the FANTASTICO Lifestyle questionnaire in the methods section. Specifically, we highlight its comprehensive nature in assessing multiple lifestyle domains and its applicability to a student population and it is already validated to the Portuguese population.

RESULTS

The authors report gender differences in mental health in association with lifestyle factors. Differences were favorable to male participants compared to women. I recommend adding a specific paragraph about gender differences.

Response: We have now added a dedicated paragraph to the results section discussing gender differences in mental health and lifestyle factors, providing additional detail and interpretation of the findings.

DISCUSSION

Line 397. The authors discuss that women exhibited healthier lifestyle compared to men. These findings are based on the general population, but not in clinical populations (diabetes mellitus, people with mental health disorders). I recommend to add some words about clinical populations, to discuss if these gender differences apply to people with health problems, and are or not in the same line that findings from students' samples.

Response: We have expanded this discussion to include clinical populations, addressing whether these gender differences hold true in individuals with health conditions such as diabetes mellitus or mental health disorders. We have also added two references discussing these differences in clinical populations.

Reviewer 2 Report

Comments and Suggestions for Authors

The manuscript titled “Mental Health and Lifestyle Factors Among Higher Education Students: A Cross-Sectional Study” provides a comprehensive analysis of the mental health and lifestyle factors among higher education students at the Polytechnic of Porto. The study is well-structured and addresses an important topic, but there are several areas where improvements can be made to enhance the clarity, depth, and overall quality of the manuscript. Below are specific comments and suggestions for the authors:

1. The introduction provides a good overview of the mental health challenges faced by higher education students. However, it could benefit from a more detailed discussion of the specific gaps in the literature that this study aims to address. For example, while the interplay between mental health and lifestyle factors is mentioned, the authors could elaborate on why this particular combination of factors (e.g., physical activity, sleep, nutrition) was chosen and how they interact in the context of higher education.

2. The objective of the study is stated as “to describe the lifestyle and levels of mental health of higher education students and determine their predictors and intercorrelations”. This could be more specific. For instance, are there particular lifestyle factors that are hypothesized to have a stronger impact on mental health? Clarifying the hypotheses or research questions would provide a clearer direction for the study.

3. The sample size calculation is mentioned, but it would be helpful to provide more details on how the margin of error and reliability level were chosen. Additionally, the authors should discuss whether the final sample size (745 students) was sufficient to detect the expected effects, especially given the diversity of the student population.

4. The description of the instruments (DASS-21, CORE-18, and FANTASTICO) is thorough, but the authors could provide more context on why these specific instruments were chosen over others. For example, were these instruments validated in similar populations (e.g., Portuguese higher education students)? If so, this should be mentioned.

5. "Actually, even though the authors used existing instruments, they still need to test their validity and reliability. This is necessary because the instruments are being used in a new context or culture. I strongly suggest that the authors cite and follow established methods to clarify the validity and reliability of these instruments:

https://doi.org/10.1002/tea.21899

https://doi.org/10.1007/s10639-022-11277-0

6. The statistical analysis section is detailed, but it would benefit from a clearer explanation of the rationale behind the chosen statistical tests. For example, why was the stepwise method used for multiple linear regression? Additionally, the authors should discuss how they handled potential confounding variables, such as socioeconomic status or prior mental health history, which could influence the results.

7. I believe that certain tables, such as Table 4, may not be necessary. This table appears to be excessively long and could be streamlined for better clarity. The authors might consider summarizing the key results in the text instead of presenting them in such an extensive tabular format.

8. The results section is comprehensive, but some findings could be presented more clearly. For instance, the authors mention that “being male was associated with better outcomes on the DASS-21 and CORE-18”, but it would be helpful to provide more context on what these “better outcomes” entail. Are males less likely to report symptoms of depression, anxiety, or stress? Providing more specific details would enhance the interpretability of the results.

9. The discussion does a good job of contextualizing the findings within the broader literature, but it could go further in explaining why certain patterns emerged. For example, why do women report worse mental health outcomes but engage in healthier lifestyle behaviors compared to men? The authors could explore potential cultural or societal factors that might explain these differences.

10. The limitations section is thorough, but the authors could expand on the potential impact of these limitations on the study's findings. For example, how might the cross-sectional design affect the ability to draw causal inferences? Additionally, the authors mention the risk of common method bias due to the use of self-report measures. They could discuss strategies to mitigate this bias in future studies, such as incorporating objective measures of mental health or lifestyle factors.

11. The authors briefly mention the need for targeted interventions, but this section could be expanded. What specific interventions might be effective based on the study's findings? For example, could universities implement programs to promote physical activity or improve sleep hygiene among students? Providing concrete recommendations would make the study more actionable for policymakers and practitioners.

12. The conclusion provides a good summary of the study’s main findings, but it could be more concise. The authors should focus on the most important takeaways and avoid repeating details that were already discussed in the results and discussion sections.

13. The conclusion briefly mentions the need for more diverse samples and additional variables, but it could be more specific. For example, what specific variables (e.g., social support, economic pressures) should future studies explore? Additionally, the authors could suggest longitudinal designs to better understand the causal relationships between lifestyle factors and mental health.

14. The manuscript is generally well-written, but there are some areas where the language could be more concise or clarified. For example, the phrase “mild levels of depression, anxiety, and stress, nearing the moderate threshold” could be rephrased for clarity. Additionally, the authors should ensure that all abbreviations are defined upon first use (e.g., DASS-21, CORE-18).

15. The tables are well-organized, but some of the headers could be more descriptive. For example, Table 2 could include a brief explanation of what each subscale measures (e.g., CORE-18_swe, CORE-18_ps) to help readers interpret the results more easily.

16. I believe that the authors should provide a clearer and more comprehensive clarification of their contributions, particularly in terms of their theoretical advancements. The theoretical framework is a crucial foundation for any research, and elucidating the specific ways in which this study extends existing theories or proposes new ones would significantly enhance the value and impact of their work. Additionally, it would be beneficial for the authors to explicitly state how their findings fill gaps in the current literature and contribute to the broader understanding of the subject matter. This level of transparency and specificity would not only strengthen the credibility of their research but also provide a clearer roadmap for future scholars in the field.

Comments on the Quality of English Language

The English could be improved to more clearly express the research.

Author Response

The manuscript titled “Mental Health and Lifestyle Factors Among Higher Education Students: A Cross-Sectional Study” provides a comprehensive analysis of the mental health and lifestyle factors among higher education students at the Polytechnic of Porto. The study is well-structured and addresses an important topic, but there are several areas where improvements can be made to enhance the clarity, depth, and overall quality of the manuscript. Below are specific comments and suggestions for the authors.

Response: We would like to sincerely thank you for your time and effort in reviewing our manuscript. We appreciate your insightful comments and suggestions, which have helped us improve the quality and clarity of our work. Below, we provide a point-by-point response to your comments and outline the revisions made accordingly. All changes in the manuscript are highlighted in blue.

  1. The introduction provides a good overview of the mental health challenges faced by higher education students. However, it could benefit from a more detailed discussion of the specific gaps in the literature that this study aims to address. For example, while the interplay between mental health and lifestyle factors is mentioned, the authors could elaborate on why this particular combination of factors (e.g., physical activity, sleep, nutrition) was chosen and how they interact in the context of higher education.

Response: We believed that despite the growing recognition of these issues, a need for more comprehensive research remains if we are to fully understand the complex relationships between various lifestyle factors and mental health outcomes among university students. Existing studies often focus on single aspects of lifestyle in isolation, overlooking the potential interactions between different factors which we intend to do in our study.

  1. The objective of the study is stated as “to describe the lifestyle and levels of mental health of higher education students and determine their predictors and intercorrelations”. This could be more specific. For instance, are there particular lifestyle factors that are hypothesized to have a stronger impact on mental health? Clarifying the hypotheses or research questions would provide a clearer direction for the study.

Response: We have refined the objective statement to specify which lifestyle factors are hypothesized to have a stronger impact on mental health. This clarification helps provide a clearer direction for the study.

  1. The sample size calculation is mentioned, but it would be helpful to provide more details on how the margin of error and reliability level were chosen. Additionally, the authors should discuss whether the final sample size (745 students) was sufficient to detect the expected effects, especially given the diversity of the student population.

Response: The sample size was calculated for a margin of error of 5% and a reliability of 90%. From a population of 21.000 Polytechnic of Porto students, a total of 378 participants were required, according to the RAOSOFT Sample Size Calculator (2004). We have added an explanation of how the margin of error and reliability level were determined. Furthermore, we discuss the adequacy of the final sample size (745 students) in detecting expected effects given the student population's diversity. We’ve added this information on the manuscript: “The sample size was calculated for a margin of error of 5% (because it balances precision and practicality) and a reliability of 95% (because minimizes Type I errors while maintaining reasonable certainty). These statistical standards are widely accepted in research and policy-making as standard thresholds (Freedman, Pisani, & Purves, 2007). From a population of 21.000 Polytechnic of Porto students, a total of 378 participants were required, according to the RAOSOFT Sample Size Calculator (2004). A statistical power analysis was conducted to confirm that 745 students were sufficient to detect the expected effects and a value of 0,95 was obtained, confirming its adequacy (Tabachnick & Fidell, 2013).

  1. The description of the instruments (DASS-21, CORE-18, and FANTASTICO) is thorough, but the authors could provide more context on why these specific instruments were chosen over others. For example, were these instruments validated in similar populations (e.g., Portuguese higher education students)? If so, this should be mentioned.

Response: We now provide additional context on why DASS-21, CORE-18, and FANTASTICO were chosen over other instruments. Specifically, we highlight whether these instruments were validated in similar populations, including Portuguese higher education students. FANTASTICO was chosen due to its comprehensive nature in assessing multiple lifestyle domains and its applicability to student populations; DASS-21 is a widely used and validated tool that could provide a reliable assessment of psychological distress, which is essential for understanding the mental health challenges faced by higher education students; CORE-18 offers insights into the emotional and mental health status of students beyond diagnostic categories, allowing for a more holistic evaluation of well-being.

  1. "Actually, even though the authors used existing instruments, they still need to test their validity and reliability. This is necessary because the instruments are being used in a new context or culture. I strongly suggest that the authors cite and follow established methods to clarify the validity and reliability of these instruments:

https://doi.org/10.1002/tea.21899

https://doi.org/10.1007/s10639-022-11277-0

Response: We acknowledge the importance of testing validity and reliability in a new cultural context. We have cited and followed established methods to clarify the validity and reliability of these instruments, as suggested. We also tested their validity and reported the results, as suggested.

  1. The statistical analysis section is detailed, but it would benefit from a clearer explanation of the rationale behind the chosen statistical tests. For example, why was the stepwise method used for multiple linear regression? Additionally, the authors should discuss how they handled potential confounding variables, such as socioeconomic status or prior mental health history, which could influence the results.

Response: We have elaborated on the rationale behind using the stepwise method for multiple linear regression. Additionally, we discuss how potential confounding variables (e.g., socioeconomic status, prior mental health history) were managed in the analysis. Potential confounding variables, such as socioeconomic status or prior mental health history, which could influence the results, were included in linear regression models. However, none of them showed any influence on the response variable, so the best models for each of the regressions performed were those reported, and none of them included potentially confounding variables.

  1. I believe that certain tables, such as Table 4, may not be necessary. This table appears to be excessively long and could be streamlined for better clarity. The authors might consider summarizing the key results in the text instead of presenting them in such an extensive tabular format.

Response: Thank you for your suggestion; key results are now summarized in the text where appropriate.

  1. The results section is comprehensive, but some findings could be presented more clearly. For instance, the authors mention that “being male was associated with better outcomes on the DASS-21 and CORE-18”, but it would be helpful to provide more context on what these “better outcomes” entail. Are males less likely to report symptoms of depression, anxiety, or stress? Providing more specific details would enhance the interpretability of the results.

Response: We have revised the results section to specify whether males (and other variables) report fewer symptoms of depression, anxiety, or stress. This clarification enhances the interpretability of the findings.

  1. The discussion does a good job of contextualizing the findings within the broader literature, but it could go further in explaining why certain patterns emerged. For example, why do women report worse mental health outcomes but engage in healthier lifestyle behaviors compared to men? The authors could explore potential cultural or societal factors that might explain these differences.

Response: We have expanded the discussion to explore cultural and societal factors that may explain why women report worse mental health outcomes but engage in healthier lifestyle behaviors.

  1. The limitations section is thorough, but the authors could expand on the potential impact of these limitations on the study's findings. For example, how might the cross-sectional design affect the ability to draw causal inferences? Additionally, the authors mention the risk of common method bias due to the use of self-report measures. They could discuss strategies to mitigate this bias in future studies, such as incorporating objective measures of mental health or lifestyle factors.

Response: We have elaborated on how the cross-sectional design affects causal inferences and discussed strategies to mitigate common method bias in future research.

  1. The authors briefly mention the need for targeted interventions, but this section could be expanded. What specific interventions might be effective based on the study's findings? For example, could universities implement programs to promote physical activity or improve sleep hygiene among students? Providing concrete recommendations would make the study more actionable for policymakers and practitioners

Response: We now provide specific intervention recommendations, such as lifestyle redesign programs.

  1. The conclusion provides a good summary of the study’s main findings, but it could be more concise. The authors should focus on the most important takeaways and avoid repeating details that were already discussed in the results and discussion sections.

Response: The conclusion has been refined to focus on key takeaways and avoid redundancy.

  1. The conclusion briefly mentions the need for more diverse samples and additional variables, but it could be more specific. For example, what specific variables (e.g., social support, economic pressures) should future studies explore? Additionally, the authors could suggest longitudinal designs to better understand the causal relationships between lifestyle factors and mental health.

Response: We now outline additional variables (e.g., social support, economic pressures) for future studies and suggest longitudinal designs to better understand causal relationships.

  1. The manuscript is generally well-written, but there are some areas where the language could be more concise or clarified. For example, the phrase “mild levels of depression, anxiety, and stress, nearing the moderate threshold” could be rephrased for clarity. Additionally, the authors should ensure that all abbreviations are defined upon first use (e.g., DASS-21, CORE-18).

Response: We have revised certain phrases for improved clarity and ensured that all abbreviations are defined upon first use.

  1. The tables are well-organized, but some of the headers could be more descriptive. For example, Table 2 could include a brief explanation of what each subscale measures (e.g., CORE-18_swe, CORE-18_ps) to help readers interpret the results more easily.

Response: We have revised table headers to be more descriptive and included brief explanations to aid interpretation.

  1. I believe that the authors should provide a clearer and more comprehensive clarification of their contributions, particularly in terms of their theoretical advancements. The theoretical framework is a crucial foundation for any research, and elucidating the specific ways in which this study extends existing theories or proposes new ones would significantly enhance the value and impact of their work. Additionally, it would be beneficial for the authors to explicitly state how their findings fill gaps in the current literature and contribute to the broader understanding of the subject matter. This level of transparency and specificity would not only strengthen the credibility of their research but also provide a clearer roadmap for future scholars in the field.

Response: We have explicitly stated how our findings contribute to existing theories and fill gaps in the literature. Additionally, we highlight how this study extends current theoretical frameworks and provides a foundation for future research.

Round 2

Reviewer 2 Report

Comments and Suggestions for Authors

I do not have further comment.

Comments on the Quality of English Language

I do not have further comment.